# Inter-Task Transfer of Prism Adaptation through Motor Imagery [note 1]

**DOI:** 10.3390/brainsci13010114

**Published:** 2023-01-09

**Authors:** Lisa Fleury, Léa Dreyer, Rola El Makkaoui, Elise Leroy, Yves Rossetti, Christian Collet

**Affiliations:** 1INSERM UMR-S U1028, CNRS UMS 5292, Trajectoires Lyon Neuroscience Research Center (CRNL), 69500 Bron, France. Claude Bernard University of Lyon 1, 69100 Villeurbanne, France; 2Inter-University Laboratory of Human Movement Biology (EA 7424), Claude Bernard University of Lyon 1, 69100 Villeurbanne, France; 3Defitech Chair of Clinical Neuroengineering, Neuro-X Institute (INX) and Brain Mind Institute (BMI), École Polytechnique Fédérale de Lausanne (EPFL) Valais, 1950 Sion, Switzerland; 4Department of Psychology, University of Lyon 2, 69007 Lyon, France; 5“Mouvement et Handicap” Platform, Neurological Hospital, Hospices Civils de Lyon, 69500 Bron, France

**Keywords:** prism adaptation, motor imagery, transfer, after-effects, sensory realignment, internal models

## Abstract

Prism adaptation (PA) is a useful method to investigate short-term sensorimotor plasticity. Following active exposure to prisms, individuals show consistent after-effects, probing that they have adapted to the perturbation. Whether after-effects are transferable to another task or remain specific to the task performed under exposure, represents a crucial interest to understand the adaptive processes at work. Motor imagery (MI, i.e., the mental representation of an action without any concomitant execution) offers an original opportunity to investigate the role of cognitive aspects of motor command preparation disregarding actual sensory and motor information related to its execution. The aim of the study was to test whether prism adaptation through MI led to transferable after-effects. Forty-four healthy volunteers were exposed to a rightward prismatic deviation while performing actual (Active group) versus imagined (MI group) pointing movements, or while being inactive (inactive group). Upon prisms removal, in the MI group, only participants with the highest MI abilities (MI+ group) showed consistent after-effects on pointing and, crucially, a significant transfer to throwing. This was not observed in participants with lower MI abilities and in the inactive group. However, a direct comparison of pointing after-effects and transfer to throwing between MI+ and the control inactive group did not show any significant difference. Although this interpretation requires caution, these findings suggest that exposure to intersensory conflict might be responsible for sensory realignment during prism adaptation which could be transferred to another task. This study paves the way for further investigations into MI’s potential to develop robust sensorimotor adaptation.

## 1. Introduction

Sensorimotor plasticity allows human beings to set up adaptive processes to keep movement accuracy and smoothness when facing sustained perturbations or new situations. The processes involved to deal with such perturbations can be reflected by transfer properties, i.e., the capacity to apply sensorimotor transformations from the initial perturbation context to other situations [1,2]. In addition, the understanding of such transfer processes represents a crucial issue in the field of neuro-rehabilitation [3].

Prism adaptation (PA) is one of the oldest paradigms (first described by von Helmotlz in 1962) aimed at investigating short-term sensorimotor plasticity processes (for a review, see [4]). Individuals’ baseline pointing performances are first assessed. Subsequently, individuals are actively exposed to a visual field shift through prismatic goggles. During the first following pointing trials, they experience pointing errors in the direction of the prismatic deviation. Over a couple of trials, the participants rapidly compensate for errors until they return to their baseline performance. Once prisms are removed at the end of exposure, they make pointing errors in the direction opposite to the initial prismatic deviation, called “negative after-effects”. The proof of a “true” adaptation is attested by the after-effects assessment, relative to the baseline performance [5,6].

Several interpretative frameworks have been proposed to describe processes at work during PA. Redding and Wallace [7] supposed two distinct processes, strategic control (recalibration), and sensory realignment. Strategic control rapidly reduces errors as soon as the exposure begins, by means of strategic adjustments of motor plans. During the very first trials, the participants use previous errors to correct the subsequent pointing trial using cognitive–compensatory modifications of motor commands, even, for example, voluntary pointing-off target strategies. Realignment is a slower and automatic process needed to reorganise the visual and proprioceptive reference frames. Visual information is distorted by the prismatic lenses while proprioceptive information remains unaltered, resulting in a mismatch between the two types of sensory feedback. Realignment aims at reducing sensory conflict by updating relationships between visual and proprioceptive reference frames. After-effects originating from realignment are considered “true” adaptation [4,5,8].

However, disentangling both strategic control and realignment and their specific contributions to prism adaptation is still not fully elucidated. The conditions required to develop after-effects have been deeply investigated but remained somewhat unclear.

Over the building of PA literature knowledge, conscious error detection and/or correction during prism exposure are necessary for adaptation [9]. Moreover, it has been reported that visual feedback was needed to develop adaptation and could not be replaced by auditory–verbal feedback [10]. Recent pieces of work challenged this assumption concerning the conscious detection of an error, for example, by developing adaptation with a multiple-step paradigm, i.e., gradually inducing prismatic deviation by steps of two degrees [11]. Therefore, participants remained unaware of the prismatic perturbation and did not detect errors although they showed substantial after-effects once prisms were removed [11,12,13,14,15].

Another stronger assumption was that active movements are needed to develop PA, in order to generate actual sensory feedback [5,9,16] although passive (robotic-guided) movements have also been associated with after-effects following exposure to a visual shift in a virtual environment [17] or visuomotor rotation [18]. However, strict passive exposure (i.e., absence of movement) did not elicit after-effects [5]. Therefore, whether motor imagery (MI) is likely to elicit after-effects should be questioned.

MI is the mental representation of an action without any concomitant execution, shares temporal features with actual movement [19], and activates common cortical areas [20,21]. This neuro-equivalence is correlated with MI ability, i.e., the capacity to easily form vivid mental representations of movement. Good imagers demonstrate stronger activation of motor areas during MI compared to poor imagers [22]. Importantly, MI has the potential to improve actual motor performance through mental training [23,24]. In addition, Michel et al. [25] demonstrated that MI had the potential to develop motor adaptation when exposed to prisms. In fact, participants who performed pointing movements through MI (i.e., with no actual execution of pointing movements) while exposed to a 15 degrees prismatic deviation showed significant after-effects following exposure. These were nevertheless (less-than-half) smaller compared to those following the actual pointing. Crucially, no after-effect occurred in the control groups. The control group data showed that in the prisms-imagery group, after-effects were neither attributable to (1) simple exposure to prisms (without MI, with or without focused attention on hand or targets), nor to (2) simple practice of MI (without wearing prisms). Importantly, the last control group probed that viewing the hand while imaging movements under prism exposure was required to develop after-effects. In fact, participants who performed pointing movements through MI under prisms exposure but with no vision of their hand did not show any adaptation. Thus, sensorimotor conflict (i.e., the mismatch between predicted and actual consequences of movements) was not required to develop adaptation. However, the inter-sensory conflict between biased visual information and unbiased proprioceptive information was strong enough to produce after-effects.

A possible interpretation relies on the internal model’s framework [26,27], as MI involves the internal forward model [28]. It predicts the future sensorimotor state through the expected consequences of motor execution, by integrating the efferent copy of the motor command and the current state of the sensorimotor system [27]. In this case, while imagining pointing movements under prism exposure, the participants integrated the efferent copy of their successful pointing movement as well as the initial sensory information, i.e., the visually perceived position of the target and the perception of their hand through visual and proprioceptive information. Both target and hand positions are biased by prisms, while the proprioceptively-perceived hand position is not. This inter-sensory conflict may lead participants to adopt an intermediary hand position between the visual shifted and the proprioceptive stable hand locations. Through the repetition of MI pointing trials, participants gradually realigned the proprioceptive hand location toward the target, thus eliciting after-effects. A complementary experiment showed that proprioception was realigned with vision [25]. Crucially, the process of strategic control was precluded in this study because no error correction was possible. Thus, sensorimotor predictions alone have the potential to update internal models and generate adaptation (through sensory realignment) as long as the inter-sensory conflict goes on [25]. This finding was recently supported by the study of motor imagery under microgravity, which represents another type of sensorimotor perturbation [29]. Alterations in gravity induce adaptive changes in sensorimotor control (for a review, see [30]). Rannaud-Monany et al. [29] showed that the flow of sensory information perceived by the subject in a microgravity environment coupled with motor imagery of arm pendular movements elicited adaptation as assessed by changes in imagined movement duration. However, adaptation was not observed following mere exposure to microgravity without motor imagery.

Noteworthy, motor imagery represents the opportunity to study motor cognition while disregarding actual sensory feedbacks. Nonetheless, although previous experiments determined which conditions are likely to trigger after-effects and sensorimotor adaptation [4,8], nothing is known about the extent to which after-effects elicited through MI pointing are transferable to other tasks.

An important feature of PA methodology is the need to assess after-effects using a different context as to that of the exposure (without prismatic goggles and toward different targets). This is the condition to attest to “true” adaptation [5,8]. Yet, testing after-effect transfer better attests to after-effects that are not specific to the context of exposure [31] and provides crucial information related to the processes involved during visual perturbation. During prism exposure, compensations could easily be measured in a different context by changing target locations [32,33], limbs [34,35,36], or even tasks [37,38]. We recently tested how after-effects transferred between throwing and pointing in PA [31]. The first group of participants was exposed to the prismatic deviation while performing throwing, with the aim to test how after-effects transferred from throwing to pointing. Another group tested how after-effects developed during pointing prism exposure transferred to throwing. We reported a single transfer from pointing to throwing but not the other way around. However, participants with expertise in throwing exhibited reduced variability and showed transfer to pointing. Conversely, high variability in throwing trajectories in the novice group may explain the absence of transfer from throwing to pointing. This study revealed that transfer capacities may uncover the contribution of different processes at work when facing a similar perturbation. Therefore, transferable after-effects may imply “true” adaption due to sensory realignment. However, understanding the processes underlying inter-task transfer needs additional data.

The purpose of this study is to provide additional knowledge about the conditions needed to develop transferable after-effects by investigating whether MI has the potential to induce short-term plasticity when facing visuomotor perturbation. Therefore, the aims of this work are: (1) to demonstrate that MI during prism exposure can induce adaptation, with reference to a previous study by Michel et al. [25], (2) to test whether adaptation through MI can be transferred to another task; (3) to test to what extent MI abilities can affect MI-based adaptation and transfer.”

## 2. Materials and Methods

### 2.1. Participants

Forty-four right-handed participants with normal vision took part in the present study. The sample size was defined on the basis of a previous study investigating PA by mental practice [25], to achieve a minimal number of 10 participants for each experimental group. None have ever been exposed to prisms and all were naive about the paradigm. They had no history of neurological conditions nor right upper limb orthopaedic trouble. All participants gave their written informed consent before starting the protocol. All procedures were conducted in accordance with relevant guidelines and were approved by an ethics committee (CEEI (Ethics Evaluation Committee of Inserm) n°19-578; IRB00003888).

All participants were first assessed for their MI ability using a revised version of the movement imagery questionnaire (MIQ-R, [39]) before being distributed into three groups “Active” (ACT), “Inactive” (INA), and “Motor Imagery” (MI). This questionnaire is made of 4 movements involving the upper limbs through visual and kinaesthetic (i.e., proprioceptive) imagery (8 items in total). The experimenter demonstrated each movement before the participant was asked to reproduce it once, and then performed the mental representation of the movement either visually (by mentally recalling visual information usually associated with the actual execution of the same movement) or kinaesthetically (by mentally recalling static and dynamic information from our own body in motion). Kinaesthetic or proprioceptive MI is the representation of body segment positions as well as their movements usually associated with the actual characteristics of action execution, i.e., the sense of movement). Thereafter, she or he had to self-report the ease of mentally representing the action on a 7-point Likert scale, from 1 (very hard to see/feel) to 7 (very easy to see/feel). To ensure minimal MI abilities for all subjects, we planned to exclude participants with an average MIQ-R score below 4, corresponding to the neutral item of the scale (no easy nor hard to see/to feel). Then, participants were pseudo-randomly assigned to the active (“ACT”; n = 13, mean MIQ-R = 5.48 ± 1,08), the inactive (“INA”; n = 11, mean MIQ-R = 5.78 ± 0,89) or the Motor Imagery (“MI”; n = 20, mean MIQ-R = 6.08 ± 0.52) group. The MI group was then subdivided into 2 subgroups depending on their MI ability and based on a median split: the top 10 participants with the highest MI ability (i.e., good imagers, mean = 6.51 ± 0.41) were assigned to the “Motor Imagery+” (MI+) group while the 10 others with lower scores (i.e., lower imagers, mean = 5.65 ± 0.59) were assigned to the “Motor Imagery−“(MI−) group.

### 2.2. Experimental Procedure

Participants followed the four same stages as shown in Figure 1: pointing («exposed» task) and throwing («unexposed» task). Tasks were performed either with vision (closed-loop) or without (open-loop), and toward a central target (see the next section for details). Specific conditions are detailed hereafter. Familiarization, pre-tests, and post-tests were similar for all participants, whatever their group. Both tasks were performed in a pseudo-randomised order. Exposure conditions varied depending on the group to which participants were assigned.

#### 2.2.1. Familiarization

To familiarize themselves with both tasks and experimental settings, the participants of each group performed 30 throwing movements and 30 pointing movements (6 sessions of 5 pointing movements, as detailed in the Tasks section). All trials were performed in closed-loop conditions, with the vision toward the central target. Each sequence of 5 successive pointing movements was timed to be compared to that of MI pointing.

#### 2.2.2. Pre-Tests

Baseline performances (i.e., terminal errors) were assessed on each task during pre-tests. Participants in both groups performed 20 successive throwing movements and 20 pointing movements, in open-loop conditions (no vision).

#### 2.2.3. Exposure

All participants wore prismatic goggles that laterally shifted the visual field 10 degrees toward the right (OptiquePeter.com, Lyon). Before wearing the goggles, participants were asked to keep their eyes closed. While wearing the prisms, they were also instructed not to look at their own body or move any limb (except when performing the task). Before exposure, each participant’s right index finger was placed in the starting position (see Figure 2). All participants saw their right hand and target simultaneously.

In the active group, participants performed 20 actual sequences of pointing (100 pointing movements in total) as fast and accurately as possible, in closed-loop conditions (i.e., vision of the hand position was continuously allowed during movement). Participants should act after a vocal signal provided by the investigator.

In the MI group, participants performed 20 sequences of mental representations of pointing. They should imagine themselves performing accurate pointing sequences toward the different targets, with both kinaesthetic and visual information. They were allowed to look at either their hand or the targets through the prisms and were asked to not look at other parts of their environments. Every 5 sequences, participants were asked to self-report MI vividness on a 5-point Likert scale. More precisely, they should report the intensity/clarity of their mental representations from 1 (no sensation/no image at all) to 5 (sensation as intense as during actual execution/images as clear as during watching a video). Additionally, each pointing sequence was timed to be compared with those from actual pointing during familiarization.

In the inactive group, participants saw both their hand and targets through the goggles and should watch their motionless hand with their index placed in the starting position, for 5 min. They were instructed to remain motionless, without looking at any part of the environment. They did not receive any other information. To prevent participants from imagining any movements, the investigator asked them trivial questions, so they did not deal with any mental material during the experiment. At the end of exposure, all participants kept their eyes closed before removing their goggles.

#### 2.2.4. Post-Tests

Once the prisms were removed, after-effects were assessed in both tasks under open-loop conditions (no vision allowed). Participants performed 20 actual throwing movements and 20 actual pointing movements toward the central target.

### 2.3. Tasks

Each participant sat in an adjustable chair in front of the pointing desk during the whole experiment. The participants also had to keep this position with their heads resting in V-shaped chin rests. During each transitional phase, they were asked to close their eyes to be deprived of visual information (thus, preventing visuomotor compensations). The next section details the experimental set-up of each task, as illustrated in Figure 2.

#### 2.3.1. Pointing Task

The starting position of the finger was lined-up with the body midline either below the chin rest (out of view; pre-tests and post-tests) or a few centimetres ahead, i.e., within the visual field (visible; familiarization and exposure-Figure 2). Five targets were horizontally placed on the pointing desk (Figure 2). Participants had to perform two types of pointing movements, depending on the step of the experiment: (i) direct pointing sequence to the five targets, and (ii) unique straightforward pointing directed to a central target.

Sequences of pointing movements (familiarization and exposure)

Full vision allowed the participants to see both their hands and the targets. To complete the sequence, participants had to reach each target from target 1 to target 5 (see Figure 2) as quickly and accurately as possible, while going back to the starting position between each target. Each sequence was made of 5 pointing movements (from starting position to each target) with no interruption. The starting signal was given by the experimenter before each sequence. During familiarization, all participants performed actual movements. During exposure, only participants from the active group performed actual movements while those of the MI groups imagined movements. Finally, participants in the inactive group neither executed nor imagined any movement during exposure. Sequential pointing provided sequences of appropriate duration to perform vivid MI as the rule of isochrony between the actual movement and its mental representation only works for a movement duration of about ten seconds (above and below this average value, mental representation duration is under- and overestimated, respectively (for a review, see [19]).

Unique straightforward pointing movements (pre and post-tests).

During this type of pointing, no vision was allowed during the entire movement and participants had to produce a unique and straightforward movement toward the central target (target n°3, see Figure 2, left panel). To control visual feedback, the experimenter manipulated a cover board in front of the participants. Before each trial, she lowered the cover board, so that the participants saw the targets while their index was still invisible in the starting position below the chin rest. Before each trial, the experimenter lifted the cover board to prevent target vision and the forthcoming movement. This procedure allows the reliable evaluation of after-effects and prevents de-adaptation [5,8]. During pre-tests and post-tests, all participants followed the same pointing procedure, irrespective of their group.

#### 2.3.2. Throwing Task

Participants were in the same positions as during the pointing task, with their heads in the chin rests. The central target on which the participants had to throw a ball was placed on a vertical board, 2 m in front of the participants at the height of their eyes. The starting position requested to keep their right hand close to their ear. He or she could then throw the ball toward the central target. There were two different conditions for throwing, as for pointing.

Closed-loop throwing movements (familiarization).

The participants saw their movements and the associated results. None performed the throwing task during exposure.

Open-loop throwing movements (pre-tests and post-tests).

The starting position was similar to that in the closed-loop condition. Before each trial, participants had to hold the ball, which was placed in their right hand by the experimenter, and look at the central target. Before throwing, they were instructed to close their eyes and throw the ball without visual feedback on execution and final outcome. They could open their eyes before the next trial and, thus, see the target before each trial. We also placed a cover board to ensure that participants did not have any visual feedback during throwing.

### 2.4. Data Collection and Processing

We recorded performance, i.e., the pointing and throwing distance from the target, during pre-tests and post-tests with a high-resolution camera (4K digital camera HCVX990EF, Panasonic, Kadoma, Japan) placed between the pointing desk and the throwing board. We then processed the data with motion analysis software (Kinovea, Charmant, 2004). The software provided the lateral distance between the target and the index or the distance between the ball impact and the target. The high 4K resolution of the camera coupled with specific cautions from the Kinovea Guidelines (e.g., related to the placement of the camera; Kinovea Documentation Authors) provided optimal precision of measurement. Lateral distances were then converted into angular deviations before statistical analysis and were then considered as the main dependent variable.

### 2.5. Statistical Analysis

The dependent variable was the subject’s performance, i.e., the terminal errors on throwing and pointing before (pre-tests) and after (post-tests) exposure. First, we performed paired-sample *t*-tests independently for each group, thus comparing pre-test and post-test performances during pointing and throwing.

Then, we computed after-effects for each task as the differences between pre-test and post-test performances. We conducted two independent ANOVAs to assess the effects of Group (ACT, MI+, MI−, INA) on pointing and throwing after-effects separately. Finally, we conducted an ANOVA with repeated measures with the tasks (pointing after-effects vs. throwing after-effects) as a within-subject factor, and the group (ACT, MI+, MI−, INA) and order of the tasks during post-tests (pointing first (1) vs. throwing first (2)) as inter-subject factors.

## 3. Results

### 3.1. Comparisons between Pre-Tests and Post-Tests Performances

#### 3.1.1. Pointing Performances

Post-test pointing performances significantly shifted leftward in the ACT (t(12) = 9.50, *p* < 0.01) and MI+ (t(9) = 2.91, *p* < 0.01) groups. However, post-test pointing errors did not differ from the pre-test in MI− (t(9) = 0.89, *p* = 0.39) and INA (t(10) = 1.62, *p* = 0.14) groups (see Figure 3). PA by physical practice and mental practice for individuals with high scores of MI abilities significantly increased the negative errors in pointing while participants with lower scores of MI abilities did not show any differences in pointing performances after prism exposure compared to pre-tests.

#### 3.1.2. Throwing Performances

The statistical analysis did not reveal any difference between pre-tests and post-tests during throwing in the MI− (t(9) = 0.72, *p* = 0.52) and INA (t(10) = 0.82, *p* = 0.43) groups. Participants in the ACT group showed a non-significant leftward shift (t(12) = 1.10, *p* = 0.30). However, the difference between pre-tests and post-tests reached significance in the MI+ group (t(9) = 2.43, *p* = 0.04): throwing performance was significantly shifted leftward after PA by mental practice only for participants with high MI abilities (see Figure 4).

### 3.2. Comparisons between Groups

We performed two independent ANOVAs to assess the differences between groups related to pointing and throwing after-effects (post-tests performances with baseline subtracted, Figure 5).

For pointing after-effects, ANOVA revealed a significant effect of Group (F(3,37)=6.71, *p* < 0.01). A post hoc analysis, including correction for multiple comparisons, showed that pointing after-effects were larger in the ACT group (mean = −2.18 ± 0.9 degrees) compared to the INA (mean = −0.54 ± 1.01 degrees, *p* < 0.01) and MI− groups (mean = −0.31 ± 1.10 degrees, *p* < 0.01). The pointing after-effects in the MI+ group (mean = −1.39 ± 1.42 degrees) were larger compared to the MI− and INA groups without reaching significance.

ANOVA for transfer to throwing did not reveal any significant differences among groups (F(3,35) =0.52, *p* = 0.67, NS).

### 3.3. Task Order Influences Pointing and Throwing After-Effects

We conducted an ANOVA with repeated measures on after-effect performances with the task as a within-subject factor (pointing vs. throwing), and the group and task orders (pointing first vs. throwing first) as between-subject factors.

Task, Task*Group, Task*Task Order, Task*Group*Task Order, and Group*Task Order interaction effects did not reach significance. However, the data analysis showed a task order main effect (F(1,30) = 6.48, *p* = 0.01) showing that the average after-effects (over the group and task levels) were significantly larger when pointing after-effects were assessed first during post-tests.

A post hoc analysis corrected for multiple comparisons also showed a slight specific Task*Task order effect but no three-way interaction. Specifically, mean pointing after-effects were larger when assessed first rather than when mean throwing after-effects were assessed first (p_Tukey_ = 0.02). In addition, mean throwing after-effects were larger when assessed second rather than first (p_Tukey_ = 0.04). None reached significance when tested within each group.

## 4. Discussion

The aim of this study was to test whether prism adaptation (PA) by motor imagery (MI) of pointing elicited the transfer of after-effects to a throwing task that has not been previously experienced under prism exposure. We also investigated the effects of MI abilities on adaptation and transfer.

Data showed that post-tests in pointing significantly shifted leftward in the “MI+” and “ACT” groups after prism exposure, thus attesting to adaptation. However, post-tests in pointing were similar to pre-tests in the “MI−” and “INA” groups, showing no adaptation. Participants who were actively exposed to prisms during pointing (“ACT” group) showed substantially negative after-effects during post-tests. This is consistent with classical prism adaptation, showing that active concurrent exposure (with hand vision during the whole movement) leads to adaptation and after-effects [5]. In addition, the MI of pointing during prism exposure (without any concurrent movement) elicited a substantial leftward shift of pointing errors for participants with high MI abilities. These results replicate the findings by Michel et al. [25] and provide additional evidence that PA likely relies on sensory realignment based on inter-sensory conflict without the need for being exposed to sensorimotor conflicts [25]. It is noteworthy that exposure to prisms in the absence of MI does not produce consistent after-effects as shown by the current experiment, which is consistent with classical studies using passive exposure, e.g., [14]. The sensory realignment was ensured by the comparison of the initial hand position (computed between visual-shifted and proprioceptive unbiased locations; [40]) together with the efferent copy of pointing movements. Noticeably, participants in the “ACT” group showed a greater magnitude of after-effects although it was not twice as large as in the “MI+” group, as in Michel et al.’s study (2013). This finding is reminiscent of a more recent study demonstrating that sensory inflow from the perturbed environment (microgravity), coupled with the MI of arm movements, triggers adaptive processes without the need for action-related actual feedback [29].

As a novelty, the main objective of this study was to determine whether PA through MI transferred after-effects to throwing. Participants with high MI abilities who imagined pointing under prism exposure showed significant throwing after-effects, despite the fact that they had previously neither performed nor imagined throwing under prism exposure. Interestingly, this amount of transfer was comparable to the initial magnitude of after-effects during pointing. This result is all the more significant when considering that both participants in the MI− and INA groups during exposure did not show any significant throwing after-effects. This provides evidence that it is possible to achieve realignment following PA through MI without actual execution of movement, and that this realignment may be transferrable to another task. However, the direct comparison between throwing after-effects in the MI+ group and the control inactive group did not show any significant difference, thus suggesting that the substantial throwing after-effects in the MI+ group might not be robust enough to make any firm conclusion when compared to the control group. Although a cautious interpretation is needed, the present result still corroborates the idea that sensorimotor adaptation elicited through MI helps to update internal models [25,29]. An interesting hypothesis emerging from these findings is that the process of recalibration (which was not observed during prism exposure through MI) may favour contextual sensorimotor alterations, i.e., poor transfer. Yet, recalibration did not occur during prism exposure by MI as movements were not actually executed. Thus, we may hypothesise that the relative contribution of sensory realignment increased when recalibration was suppressed. Therefore, we suppose that sensory realignment may directly update internal models inherent to the participants’ intrinsic reference frames, thus leading to a “true” adaptation [6] and generalizing after-effects to extended motor situations [31]. This is reminiscent of several works supporting the notion of updated reference frames and internal models of spatial representation following PA by demonstrating a cognitive expansion of after-effects beyond the motor domain (for a review, see [41]). Plus, throwing is considered a ballistic movement as it does not require a deceleration phase, i.e., the release of the ball occurs at the peak of velocity [42]. Controlling the ballistic movement does not entail feedback-based online corrections but rather feedforward adjustments based on internal forward models and sensory predictions [43]. Therefore, the observation of transfer on this task supports the idea of updated internal models following PA by MI.

Another interesting and original finding relates to the difference between good and worse imagers. Participants with high MI abilities showed substantial pointing and throwing after-effects while participants with lower abilities did not. Therefore, high abilities were needed for successful PA through MI, and to transfer after-effects to the unexposed task. These differences may be explained by the relationships between MI abilities and the neuro-functional correlates of MI. Individuals with high MI capacities should recruit motor-related areas to a greater extent compared to participants with low MI abilities, suggesting a greater induced plasticity [22,44]. However, our study included participants with relevant MI abilities (average MIQ-R score of at least 4 out of 7). This relative homogeneity and the consequent split between good and worse imagers in the MI group might be controversial as worse imagers still can be considered as having relevant MI abilities. Further studies might include a broader range of MI abilities in order to deepen the relationship between MI abilities and the capacity to develop transferable sensorimotor adaptation.

Altogether, these findings are very encouraging in probing transferable adaptation by means of MI under prism exposure. Nevertheless, we did not observe any difference between the “MI+” group and both the “MI−” and “INA” groups about the magnitude of pointing and throwing after-effects, although pre- and post-tests were significantly different in MI+ but not in the two other groups. A working hypothesis is that the magnitude of pointing after-effects in the good imagers group was lower in our study than in the study by Michel et al. [25]. This could be mainly due to the difference in optical deviation, 10 degrees versus 15 degrees, respectively. In addition, the MI of pointing under prism exposure was different, as Michel et al. requested the participants to perform one hundred single pointing movements, each trial being interspersed by 5 s pauses. Although the number of trials was comparable, the exposure duration was shorter in our study than in that of Michel et al. [25]. A recent paper showed that slight differences in experimental procedures of motor adaptation can lead to inconsistent results [45]. We can hypothesise that a greater optical deviation and an extended number of sequences of pointing during exposure (increased exposure duration) may lead to an increased magnitude in pointing and throwing after-effects. Accordingly, differences between after-effects in MI+ and other groups (MI− and INA) might have been underestimated in the present study. This hypothesis awaits experimental validation.

Surprisingly, the difference between throwing pre-tests and post-tests was not significant in the ACT group. Contrary to our expectations, we did not observe the substantial transfer of after-effects from pointing to throwing for participants who performed active pointing movements under prism exposure. Although we observed increased throwing-leftward errors following exposure compared to pre-test errors, this difference did not reach significance. Yet, transfer from pointing to throwing was already demonstrated and replicated in a previous study [31]. This may be related to the divergence in exposure conditions. In the current study, active exposure was concurrent: hand vision was available from the starting position and during the whole pointing phase. Concurrent exposure may lead to reduced after-effects [46,47]. Moreover, movements were sequential and not interspersed with consistent breaks, while the participants performed six blocks of ten pointing movements in Fleury et al.’s study [31]. The total duration of exposure may have been reduced in the present study by comparison with the previous study. Altogether, these observations provide additional features related to the exposure conditions needed to observe transferable after-effects following prism exposure.

The last point of discussion is the influence of the task order during post-tests on the magnitude of pointing and throwing after-effects. We found a global task order effect during post-tests, suggesting that mean overall after-effects (averaged over the levels of groups and tasks) were larger when the exposed task’s (pointing) after-effects were first assessed. This may suggest that a potential part of the transfer is first developed by experiencing after-effects on the exposed task. In addition, mean throwing after-effects were higher when secondly assessed, when compared to throwing after-effects tested first. This corroborates the previous hypothesis. Finally, mean pointing after-effects were larger when first measured as compared to mean throwing after-effects when first measured. A logical explanation is that the after-effects on the exposed task would be greater than the after-effects on the unexposed task when both are tested immediately after exposure. Nonetheless, these effects did not influence differences among groups because task orders during pre-tests varied within each group (task order effects disappeared when tested within each group).

## 5. Conclusions

The present work demonstrated that MI of pointing movements during PA resulted in a significant adaptation that was likely transferred to another motor task (throwing) that was neither practiced nor imagined during the prismatic perturbation, only for participants with the highest MI abilities. This provides encouraging support to develop PA and transfer through MI. This study emphasises the fact that the contribution of sensory realignment (through sensory prediction error-based processes) might be related to transferable after-effects. Crucially, the perception of external errors through actual feedback of pointing movements does not seem necessary to trigger a transfer. Furthermore, the absence of execution might have increased the contribution of sensory realignment because good imagers showed a nearly whole transfer of after-effects to the throwing task. In addition, we also suggested the necessity of having high MI abilities to exhibit adaptation and transfer.

However, the absence of a significant difference between the experimental MI+ group and the control (inactive) group in terms of pointing and throwing after-effects encourages researchers to cautiously interpret the current findings. Therefore, these results will need to be confirmed and extended to larger, more heterogeneous samples, and a more powerful adaptation paradigm. In the meantime, they already pave the way for investigating sensorimotor adaptation by MI to develop transferable after-effects. The present study raises the possible use of MI in neurological rehabilitation for patients with motor alterations but preserved MI capacities [48]. For example, PA by MI could be generalised in the rehabilitation schedule of neglected patients. Future work should test how the association of MI to actual practices under sensorimotor perturbation could enhance adaptation and the transfer of sensorimotor after-effects.

## Figures and Tables

**Figure 1 brainsci-13-00114-f001:**
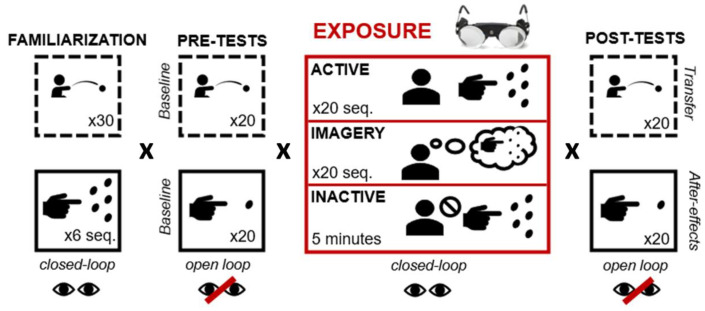
Experimental procedures included four steps: familiarization, pre-tests, exposure, and post-tests. Visual feedback was available during familiarization and post-tests and was precluded during pre- and post-tests. During familiarization, pre-tests, and post-tests, all participants performed both tasks in a pseudo-randomised order. During exposure, participants from the active group actively performed 20 sequences of pointing while those in the Imagery groups (MI+ and MI−) imagined the same 20 sequences. Participants from the inactive group remained motionless during exposure and responded to trivial questions asked by the investigator.

**Figure 2 brainsci-13-00114-f002:**
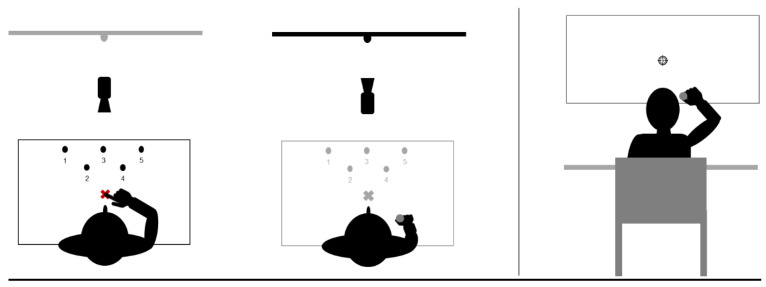
Tasks set-up. The left panel shows the top view of the pointing and throwing tasks, respectively. The right panel is a posterior view of the throwing task.

**Figure 3 brainsci-13-00114-f003:**
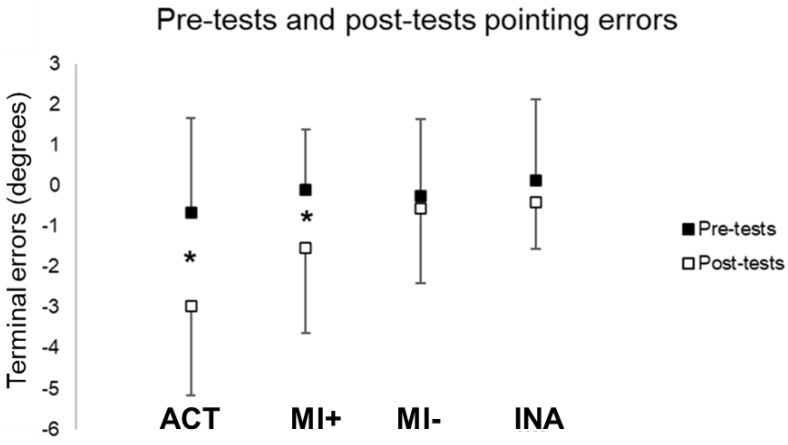
Pointing terminal errors during pre and post-tests. Values are reported with standard deviations. * means significance at *p* < 0.01. ACT: active group; MI: motor Imagery groups with higher (+) versus lower (−) MI abilities; INA: inactive group.

**Figure 4 brainsci-13-00114-f004:**
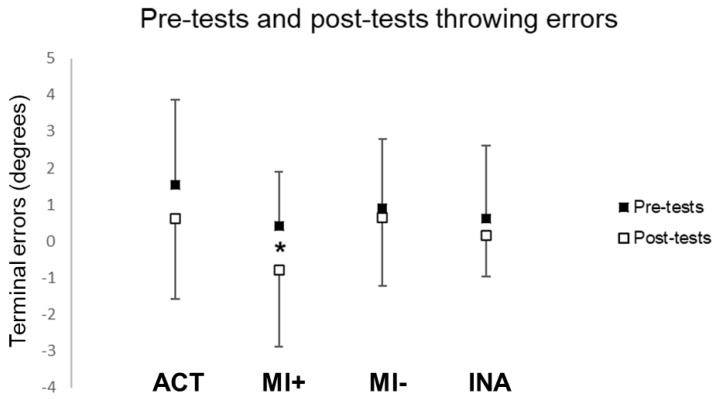
Throwing terminal errors during pre and post-tests. Values are reported with standard deviations. * means significance at *p* < 0.05. ACT: active group; MI: motor imagery groups with higher (+) versus lower (−) MI abilities; INA: inactive group.

**Figure 5 brainsci-13-00114-f005:**
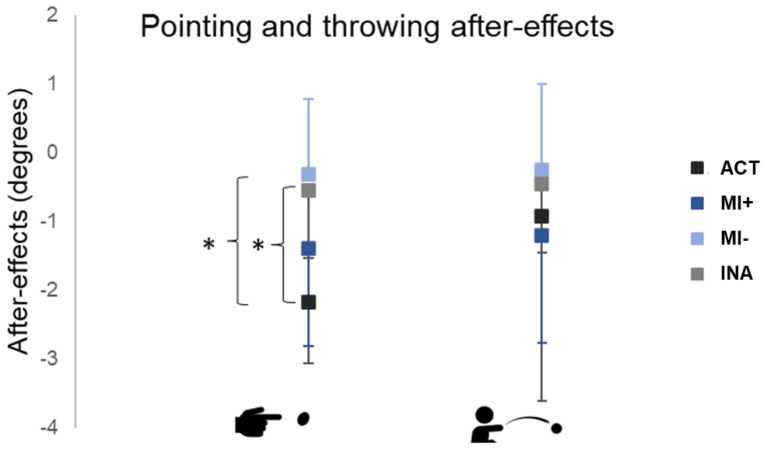
Pointing and throwing after-effects. Performances during post-tests with the baseline subtracted. Values are reported with standard deviations. * means *p* < 0.01. ACT: active group; MI: motor imagery groups with higher (+) versus lower (−) MI abilities; INA: inactive group.

## Data Availability

Data are available upon reasonable request to the corresponding author.

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
