# Peer review of "Inter-Task Transfer of Prism Adaptation through Motor Imagery†"

_brainsci, 2023, doi:10.3390/brainsci13010114_

Round 1

Reviewer 1 Report

This study investigated whether prism adaptation through motor imagery, without actual execution of movements, produces after-effects that are transferrable between tasks. The design was well-controlled and appropriate to address this question. However, the authors’ main conclusion that motor imagery does elicit transferable after-effects is based on inconsistent results – the analysis most appropriate to verify the main hypothesis actually does not support this conclusion. I believe the discussion and conclusions should be toned down according to the inconsistency of these results. Otherwise, the introduction and discussion brilliantly contextualise the research questions and the current findings within the relevant literature and highlight important theoretical implications.

1. To elaborate on the primary issue mentioned above, three types of analyses were performed, and I will focus on the first two relevant to this point. 1) Pre-test vs. post-test comparison of terminal errors within each group, within each task – this allows to verify whether a significant after-effect was present in each group and task, and indeed significant pointing after-effects were found in the ACT and MI+ groups, and significant throwing after-effect was found in MI+ group. 2) ANOVA assessing the effect of group on the after-effect in each task – I believe this analysis is most appropriate to test whether pointing after-effects transfer to throwing, and this hypothesis would be supported by demonstrating significant differences between MI and INA group in the pointing task (meaning that adaptation through MI is possible) AND in the throwing task (meaning that pointing after-effects from prism adaptation through MI do transfer to a throwing task). I imagine the purpose of having INA control group (and comparing the experimental ACT, MI+, MI- groups to it) would be to verify that the observed after-effects are robust and not only due to normal variability in participants’ performance. There was indeed substantial variability in INA group after-effects judging by the SD, and relative to that control, none of the experimental groups demonstrated a significant transfer of after-effects to the throwing task. This suggests that the significant throwing after-effect found in the MI+ group in the first analysis is not robust enough to make any firm conclusions. I believe this finding should be discussed with caution as a preliminary result (that was not confirmed by comparison to a control group), which may be replicated in a more strongly powered study (10 participants per group may not be enough to detect such variable effect) under more effective exposure conditions (which the authors already consider in the discussion).  Therefore, the title and abstract of the current manuscript are overstating the actual findings (also the last paragraph on p. 10). The results certainly suggest that it is possible to achieve realignment following prism adaptation through MI without actual execution of movements, and that this realignment may be transferrable to other tasks. The phrasing that the authors use in the discussion, that ‘these findings provide encouraging support to develop PA and transfer through MI’, would be more appropriate than the phrasing used in the abstract, which in its current state can be considered misleading and not reflecting the actual results. Especially the sentence ‘the MI group showed consistent after-effects on pointing, and, crucially, a significant transfer to throwing compared to the Inactive group’ – comparison to the Inactive group was not significant. For better accuracy, I think the authors should also specify that the main finding refers only to the MI subgroup with high MI abilities, rather than mentioning it in the next sentence (the last two sentences read as if they were conclusions based specifically on the differential MI abilities rather than observing after-effects in pointing and throwing tasks in the MI group.

2. My second concern, which can hopefully be addressed by some clarification in the methods section, relates to how participants were divided into subgroups. I understand that all participants completed the MIQ-R questionnaire, but it is not clear whether and how their scores were used to form the initial groups. Were participants selected for the MI group only if they had a score of min. 4.5, or is this number reported as a descriptive statistic? Next, how was the threshold for the high vs/ low MI ability determined? E.g., is there a standardised interpretation of MIQ scoring that creates such classification, was it a mean/median split, or were top 10 participants with the highest scores assigned to MI+ group and the rest into MI- group? The binary division into MI+/MI- seems rather arbitrary, and on that note, I wonder if it would be more informative to look at correlations between individual MI ability scores (in MI +/- subgroups combined) and the magnitude of pointing and throwing after-effects. Going back to the potential issue, if participants were pre-selected to be included in the MI group based on their minimum MI ability, the group division may be biased, e.g., if INA and ACT groups had no MI abilities. It is conceivable that MI abilities would be related to the accuracy or efficiency of generating motor predictions (involved in prism adaptation regardless of whether movement is executed or just imagined). The bottom line is that it would be useful to clarify how exactly participants were assigned to groups and what was the role of the MIQ in that, and to report the MIQ scores for each group rather than just the MI groups.

3. On a related note, was there a reason why only MI+ group reported MI vividness during prism exposure? It would seem reasonable to ask the MI- for the same ratings for better comparability. It is also possible that intermittent ratings could disrupt the adaptation process.

4. ‘Participants’ section on p. 4 should include some sample size rationale.

5. I wonder why the authors chose rightward-deviating prisms – I suspect for consistency with how prism adaptation is implemented in neglect rehabilitation (which they refer to as potential future implications in the discussion), but since this study is specifically looking at transfer effects in healthy volunteers, perhaps leftwards-shifting prism would have greater potential to induce more generalisable/transferrable effects. For instance, we know in healthy subjects adaptation to leftward-shifting prisms produces effects that transfer to visuospatial tasks whereas adaptation to rightward-shifting prisms does not. Although I appreciate that this asymmetry may be less relevant to the transfer of sensorimotor effects.

6. Please state in the statistical analysis or the results section whether any correction for multiple comparisons was applied to post-hoc contrasts (and if not, why).

Minor comments:

7. p. 1, lines 37-39: This sentence is unclear. By ‘face perturbations’ do you mean to adjust to / deal with the perturbations? What are ‘sensorimotor settings’?

8.       p. 1, line 46: ‘following’ not needed.

9.   p. 2, line 48: ‘turn back’ -> return

10.   p. 2, line 61: information is ‘biased’ -> altered/distorted/shifted?

11.   p. 2, line 63 ‘sensorial’ -> sensory

12.   p. 2, line 64: ‘After-effects originated from realignment is considered…’ -> After-effects originating from realignment are considered as evidence of “true” adaptation’

13.   p. 2, line 69: ‘over the building of PA literature knowledge’ -> literature or knowledge, both are redundant

14.   p. 2, lines 85: specify ‘mental representations’ -> of movement

15.   p. 2, lines 87-89: ‘to improve actual motor performance’ and ‘leading to motor performance enhancement’ means the same thing, the sentence is somewhat circular

16.   p. 3, 2nd paragraph: by ‘kinaesthetic’ information do you mean proprioception? It would make the introduction easier to follow if the same terminology was used throughout.

17.   p. 3, lines 117-118: ‘realigned the hand toward the target’ – by the hand, do you mean its visual / proprioceptive / visuo-proprioceptive / motor representation?

18.   p. 3, line 125: ‘changes in gravity induces’ -> induce or induced

19.   p. 3, line 126: what is meant by ‘sensory inflow’?

20.   p. 3, line 149: ‘high level’ of what?

21.   p. 4, lines 155-156: Should the sentence in square brackets be there?

22.   p. 4, line 158: ‘PA from MI’ -> PA through MI

23.   p. 4, 2nd paragraph: The statement of the study aims could be phrased more directly. It seems like there are four separate aims: 1) to replicate Michel’s study and demonstrate that MI during prism exposure can induce adaptation; 2) to test whether adaptation through MI transfer to another task; 3) to test to what extent MI abilities affect the adaptation and transfer; and 4) to elucidate the conditions necessary for these processes to occur. The last one is more generic and doesn’t seem to be directly addressed in the analyses.

24.   p. 4, line 167: ‘None have ever been exposed to prisms and [all] were naïve about the paradigm’

25.   p. 4, line 178: All participants did both, so ‘visually [and] kinaesthetically’

26.   p. 4, line 180: ‘exhibited’ -> achieved/scored

27.   p. 4, line 184: Specify “MI-“ group.

28.   Figure 1: Not clear what the large black circles mean, are they related to the order randomisation?

29.   p. 5, line 217: ‘instructed [not to look] at their own body or [move] any limb’

30.   p. 6, line 2018: ‘right index [finger]’

31.   p. 6, line 268: ‘neither executed nor performed’

32.   p. 7, section 2.4.: I may have missed it elsewhere, but this would be a good place to mention with what precision the terminal errors were measured.

33.   Figure 5: the sign of the numbers on y axis is not visible. This may also not be the best figure to represent the results of between-group comparisons as it seems to illustrate group:task interactions, especially with the points for pointing and throwing being connected. This may be misleading as the ANOVAs in this particular analysis (section 3.2 on p. 9) were done for each task separately.

34.   p. 11, lines 452-453: Not clear what the authors mean here by ‘under-evaluated’.

35.   p. 12, lines 490-491: It would be useful here to specify the direction of this effect rather than stating generic ’important role’ of MI abilities.

Reviewer 2 Report

Fleury et al., present an in-depth investigation of the mechanisms underlying prism adaptation process. The concept they present is complementary to other studies on PA and they present a nice contribution to the understanding of PA. The paper is well written and the results are straightforward. I have a few minor comments:

Introduction:

·      p. 2 Lines 79-81 – the authors state that passive exposure to prisms did not elicit aftereffects. Another relevant line of studies that should be mentioned in this context is passive adaptation through robotic devices, in which there is movement, but it is not actively initiated by the participant. In these cases it was shown that aftereffects do appear. The authors can also discuss the role of imagery in explaining these findings.
Some relevant papers:
- Cressman, E.K., Henriques, D.Y., 2010. Reach adaptation and proprioceptive recalibration following exposure to misaligned sensory input.
J. Neurophysiol. 103, 1888–1895.
- Wilf, M., Cheraka, M. C., Jeanneret, M., Ott, R., Perrin, H., Crottaz-Herbette, S., & Serino, A. (2021).
Combined virtual reality and haptic robotics induce space and movement invariant sensorimotor adaptation. Neuropsychologia, 150, 107692.

·      Another relevant finding is that adaptation through auditory-verbal feedback was not effective in inducing aftereffects:
- Bourgeois, A., Schmid, A., Turri, F., Schnider, A., & Ptak, R. (2021). Visual but not auditory-verbal feedback induces aftereffects following adaptation to virtual prisms. Frontiers in Neuroscience, 15.

·      At the end of the introduction, the authors state that their second aim is to “test the extent to which MI abilities influence adaptation and transfer processes”. Do the authors refer only to MI-based adaptation? Or also generally to ‘regular’ PA process? It is not clear at this point since the authors focus on MI-based PA in the previous phrases. Please rephrase to disambiguate and clarify this point.

Methods:

·      How was the sample size chosen? Did the authors conduct a power analysis? If not, it should be added to justify the sample size.

·      The term “kinaesthetic imagery” is not trivial and should be explained either in the methods or in the introduction. One option to explain it is state what was the exact instruction given to participants.

·      Were the participants randomly assigned into groups? Or was it based on their questionnaire scores? The description is a bit confusing. Please clarify.

·      The group MI- was not defined in the ‘participants’ section.

·      How visible was the participants’ hand during the experiment? Were they able to see the full movement trajectory? There is a difference between mechanisms of adaptation with terminal feedback or continuous feedback.

Results:

·      The authors present graphs showing the averages and standard deviations of pointing errors. Given the interindividual differences between good and bad imagers, it would also be valuable to show a scatter of individual subjects values (e.g., raincloud plots, box plots with dots for individual subjects, etc…). If pooling together all the MI group, it might be possible to tell the difference between good/bad imagers if looking only at their pointing aftereffects.

·      A related analysis that could also add to this point. Did the authors attempt to correlate the MIQ-R questionnaire results with the pointing biases?

·      Was there a difference between MI- and MI+ groups pointing results at baseline?

Discussion:

·      There are also many works that found prism aftereffect that span beyond the motor domain (see Michel 2013 for review). This is another support to the notion of updated reference frames and internal models of spatial representation following PA.
- Michel, C., 2015. Beyond the sensorimotor plasticity: cognitive expansion of prism adaptation in healthy individuals. Front. Psychol. 6, 1979.

Other comments:

·      Rephrase the term ‘Sensorimotor – motor’ in the abstract.

·      Line 367 – fix grammar
